# MANGO: Multimodal Attention-based Normalizing Flow Approach to Fusion Learning

**Thanh-Dat Truong**[1], **Christophe Bobda**[2], **Nitin Agarwal**[3,4], **Khoa Luu**[1]

[1]CVIU Lab, University of Arkansas, USA     [2]University of Florida, USA
[3]COSMOS Research Center, University of Arkansas, Little Rock, USA
[4]ICSI, University of California, Berkeley, USA
{tt032, khoaluu}@uark.edu cbobda@ece.ufl.edu, nxagarwal@ualr.edu
https://uark-cviu.github.io/projects/MANGO

## Abstract

Multimodal learning has gained much success in recent years. However, current multimodal fusion methods adopt the attention mechanism of Transformers to implicitly learn the underlying correlation of multimodal features. As a result, the multimodal model cannot capture the essential features of each modality, making it difficult to comprehend complex structures and correlations of multimodal inputs. This paper introduces a novel Multimodal Attention-based Normalizing Flow (MANGO) approach to developing explicit, interpretable, and tractable multimodal fusion learning. In particular, we propose a new Invertible Cross-Attention (ICA) layer to develop the Normalizing Flow-based Model for multimodal data. To efficiently capture the complex, underlying correlations in multimodal data in our proposed invertible cross-attention layer, we propose three new cross-attention mechanisms: Modality-to-Modality Cross-Attention (MMCA), Inter-Modality Cross-Attention (IMCA), and Learnable Inter-Modality Cross-Attention (LICA). Finally, we introduce a new Multimodal Attention-based Normalizing Flow to enable the scalability of our proposed method to high-dimensional multimodal data. Our experimental results on three different multimodal learning tasks, i.e., semantic segmentation, image-to-image translation, and movie genre classification, have illustrated the state-of-the-art (SoTA) performance of the proposed approach.

## 1 Introduction

Human perceptions interpret the surrounding world in a multimodal way via multiple input channels, such as vision, text, or audio. The deep learning-based multimodal fusion methods have majorly improved the performance of various problems, e.g., classification [20, 38, 37, 58], action recognition [12, 54, 57], semantic segmentation [65, 56, 59, 61, 60], object detection [72]. The recent large multimodal models, e.g., ChatGPT [1], Gemini [52], Chaemelon [51], LLaMMA [53], etc, introduced for general-assistant purposes have also shown impressive performance on these applications.

The critical success of multimodal fusion methods relies on the interaction and correlation mod-

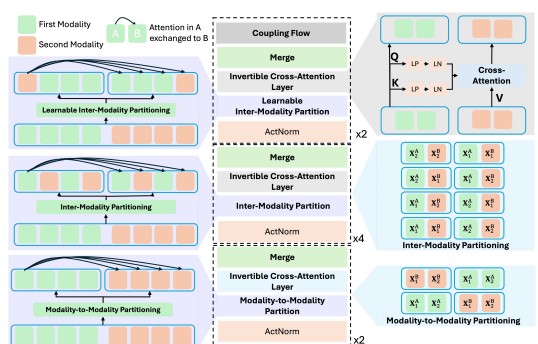

Figure 1: Our Cross-Modality Fusion Approach Via Multimodal Normalizing Flows with Invertible Cross-Attention.

39th Conference on Neural Information Processing Systems (NeurIPS 2025).

eling mechanisms across input modalities. The recent methods [20, 65] adopt the attention mechanisms of Transformers [64] to implicitly model the cross-modality correlation. By training on large-scale data, the attention models can implicitly learn the underlying correlation represented in the data. For example, the vision-language fusion models [28, 27, 48] use early fusion where the visual tokens and textual tokens are simultaneously fed into the Transformer model. Then, Transformers will learn the correlation and alignment between visual and textual tokens via the second-order correlation learning of the attention mechanism. Under this form, these multimodal fusion methods are alignment-agnostic, where the cross-modal alignments and correlations are not fully exploited [65]. In addition, the implicit fusion method often associates information across modalities without distinctly modeling the unique characteristics and correlations of each modality. Then, it may overlook the contribution of specific modalities, mainly if one modality contains more data or stronger signals, leading to suboptimal performance [65, 20]. Since the implicit approach cannot individually model the importance of each modality, these methods could struggle to capture complex structures and complementary information represented in the multimodal data. Implicit modeling methods also lack interpretability since it is hard to understand or represent the contributions of each modality to the outputs. Other methods [33, 11, 8] adopted the late fusion, where the features are fused after each of the modalities has been decided. However, late fusion ignores the low-level interaction across modalities. As a result, the direct adoption of fusion with attention could not improve performance compared to the unimodal methods [65, 20].

While most recent multimodal methods adopt attention to capture the multimodal correlations implicitly, the explicit modeling approach has been less investigated [16, 23]. The normalizing flow-based model [5, 21, 48] is a common approach to explicit modeling. By modeling the exact likelihoods of data via the bijective mapping between the data and latent spaces, the normalizing flow-based models allow for stable and reliable training, gaining better insight into the model representations of the underlying multimodal data distribution. In particular, by stacking a set of bijective transformations, the explicit models can construct complex distributions, enabling them to capture multimodal data distributions with direct control over parameters. Thus, this explicit modeling approach enhances interpretability and enables a better understanding of multimodal features and correlations in the latent space, which can be challenging to access in prior methods [65, 20]. Compared to prior methods [65, 20], explicit modeling via normalizing flows provides a better multimodal fusion mechanism since it can capture the underlying structures and correlation of multimodal data without letting any single modality dominate. Therefore, explicit modeling enables more precise, flexible, and robust multimodal fusion, improving performance in tasks requiring understanding and good alignment of multimodal data.

**The Challenges in Multimodal Normalizing Flows.** While explicit modeling is a potential approach to multimodal fusion, developing multimodal normalizing flows requires several efforts. Indeed, there are *two significant limitations in the current normalizing flow-based models*. First, while the affine coupling layer [5, 21] allows for the properties of tractability and invertibility, this layer limits the expressiveness of the models. Unlike the attention mechanism in Transformers [64], the coupling layer cannot capture the wide-range data dependencies and correlation in multimodal data [48]. Second, scaling the normalizing flow-based models to high-dimensional data is a challenging problem. It requires stacking more bijective layers in the models, leading to high computational cost and hard convergence during training [5]. While the implicit modeling approaches have alleviated the computational overhead using latent models (e.g., Latent Diffusion [42]), there are limited studies to address this overhead problem in normalizing flow-based approaches. Therefore, there is an urgent need to address these limitations to develop an efficient multimodal normalizing flow-based model.

**Contributions of this Work.** This paper introduces the new **M**ultimodal **A**ttention-based **N**ormalizing Fl**o**ws (MANGO), an explicit, interpretable, and tractable approach, to multi-modality fusion problems (Fig. 1). To the best of our knowledge, this is *one of the first studies that develops a Normalizing Flow approach to multimodal fusion learning*. Our contributions can be summarized as follows. First, we propose a new Invertible Cross-Attention (ICA) layer for Normalizing Flow-based Models. The proposed ICA layer can efficiently address the limitations of coupling layers in the standard Normalizing Flows while maintaining its tractability and invertibility properties. Second, to capture the correlation and alignment across modalities, we present three new partitioning cross-attention mechanisms, including Modality-to-Modality Cross-Attention (MMCA), Inter-Modality Cross-Attention (IMCA), and Learnable Inter-Modality Cross-Attention (LICA). Third, we present a novel Multimodal Attention-based Normalizing Flow approach with a latent model to enable its scalability to

high-dimensional multimodal data fusion. Our approach can address the limitations of computational overhead while efficiently modeling complex correlations in multimodal data. Finally, our experiments on three multimodal learning tasks, i.e., semantic segmentation, image-to-image translation, and movie genre classification, have shown the effectiveness of MANGO in different aspects, demonstrating its State-of-the-Art (SoTA) performance compared to prior multimodal models.

## 2 Related Work and Background

### 2.1 Related Work

**Attention Models.** The attention mechanism in Transformers has shown outstanding performance in unimodal and multimodal learning [64, 28, 65]. Using the second-order correlation, the attention mechanism can capture the long-term relation across input modalities. There are two common types of attention in Transformers, i.e., self-attention and cross-attention. While self-attention focuses on learning correlations within a single input modality [64], cross-attention models relationships across modalities, allowing the model to analyze complex correlations from one modality to another [68]. Transformers have become a dominant approach and have profound impacts in developing various multimodal tasks, e.g., large vision-language model [28, 27], RGB-D object segmentation [65].

**Multimodal Fusion.** Multimodal fusion learning has shown its outstanding advantage over the unimodal counterparts in various tasks, e.g., semantic segmentation [65, 20], image-to-image translation [18], action recognition [12], object detection [72], etc. The early approaches of multimodal fusion learning adopted a simple feature concatenation to fusion the information from multiple modalities [7, 76]. Then, later works further improved the cross-modality fusion by using deep fusion via a neural network, e.g., RNN [2], LSTM [50], Attention [55, 35], etc. Another approach adopted the neural architecture search to search for appropriate networks for multimodal fusion [25, 73, 10]. The current state-of-the-art fusion approaches utilize early fusion to capture cross-modality interactions at the data level via Transformers [65, 20, 28]. By combining all modalities at an initial stage via input tokens, the Transformers will learn to model the correlation across modalities via self-attention [64]. The later work further improved the early fusion method using pixel-wise fusion [20], pruning techniques [65], or dynamic multimodal fusion [70]. However, it should be noted that these current multimodal fusion methods are an implicit modeling approach.

**Explicit Modeling via Normalizing Flows.** To develop the invertible network, RealNVP [5] first introduced an affine coupling layer where its reverse version and the log-determinant of the Jacobian matrix can be easily computed. Later work [4, 21] further improved the coupling layers by introducing non-linear independent component estimation [4], invertible convolution [21, 34], activation normalization [21], autoregressive modeling [17], multi-scale architectures [5], equivariant normalizing flows [9]. Another approach [15, 48] enhanced the expressiveness of the coupling layer by using Transformers in the scaling and translation network. However, it still cannot address the problem of long-range dependencies and complex cross-modality correlation in the data. Recent studies further developed the conditional flow-based approach, e.g., conditional image synthesis [31, 32], using conditional invertible networks [47], or two invertible networks [49].

### 2.2 Limitations of Normalizing Flows

The typical normalizing flow model [5, 4, 21] is designed via the invertible affine couple layer as:

$$
\begin{aligned}
\mathbf{X}_1, \mathbf{X}_2 &= \mathrm{partition}(\mathbf{X}) \\
\mathbf{Y}_1 = \mathbf{X}_1, \quad \mathbf{Y}_2 &= \mathbf{X}_2 \odot \exp\left(\mathcal{S}(\mathbf{X}_1)\right) + \mathcal{T}(\mathbf{X}_1) \\
\mathbf{Y} &= \mathrm{merge}([\mathbf{Y}_1, \mathbf{Y}_2])
\end{aligned} \tag{1}
$$

where $\mathbf{X}$ is an input, $\mathrm{partition}$ is a partition method, e.g., RealNVP [5] adopts checkerboard partitioning method, $\mathcal{S}$ and $\mathcal{T}$ are deep neural networks, $\mathrm{merge}$ is a merging function, and $\odot$ is the element-wise matrix multiplication.

**Limitations.** The success of a flow-based model relies on the design of the invertible layers. However, the current affine coupling layers remain inefficient in modeling complex data. First, the expressiveness of the coupling layer is limited due to its simple design. The design of $\mathcal{S}$ and $\mathcal{T}$ via residual networks [5] could not capture the complex relationships represented in the high-dimensional data. Thus, it still struggles to capture highly intricate dependencies or correlations in the data,

especially in multimodal data. Second, scaling to high-dimensional data increases the complexity of the flow-based model, which can make the training unstable and inefficient. If the number of coupling layers is shallow, the model may fail to capture the complex relationships and dependencies in the multimodal data. This leads to poor performance in tasks like density estimation or fusion modeling. Thus, the high-dimensional data also requires more layers to capture all the necessary correlations among all tokens, increasing computational cost. In this paper, we will develop a new Attention-based Normalizing Flow approach to addressing these prior limitations in normalizing flows and multimodal fusion.

## 3 The Proposed Multimodal Attention-based Normalizing Flow (MANGO) Approach

Most recent multimodal models adopt Transformers with an attention mechanism to learn the cross-modality correlations [65, 20, 27]. However, prior research suggested this fusion approach is inefficient [35]. Indeed, the correlations learned via self-supervision or weak supervision cannot provide explicit attention modeling across modalities and will be ineffective when the information of multimodal inputs is sparse. In addition, as cross-modality correlations are generally high-dimensional and complex, developing a multimodal model capable of capturing complex correlations is challenging.

Therefore, to address this problem, this paper will *model the cross-modality correlations as the joint distributions*. Then, the joint distributions can be further modeled using the Normalizing Flow-based Model, a tractable yet powerful approach to modeling complex distributions with bijective mapping functions. Fig. 2 illustrates the overview of our proposed Multimodal Attention-based Normalizing Flow-based framework. Formally, let $\mathbf{X}$ be the multimodal input (e.g., RGB and Depth images), $G$ be the bijective network that maps the inputs into the latent space, i.e., $\mathbf{Z} = G(\mathbf{X})$. The prediction $\hat{\mathbf{Y}}$ can be obtained via the projection head as $\hat{\mathbf{Y}} = \mathrm{TaskHead}(\mathbf{Z})$, where $\mathrm{TaskHead}$ is the projection head that produces the task-specific outputs (e.g., semantic segmentation). Then, the multimodal data distribution $p(\mathbf{X})$ can be formed via the Normalizing Flow-based Model $G$ as in Eqn. (2).

$$p(\mathbf{X}) = \pi(\mathbf{Z}) \left| \frac{\partial G(\mathbf{X})}{\partial \mathbf{X}} \right| \tag{2}$$

where $\pi(\mathbf{Z})$ is the prior Normal distribution. In our approach, we assume that inputs $\mathbf{X}$ can be tokenized as $\mathbf{X} = [\mathbf{x}_1, ..., \mathbf{x}_N]$ where $N$ is the number of tokens. For simplicity, we assume the $\mathbf{X}$ consists of two input modalities (e.g., RGB and Depth images), i.e., $\mathbf{X} = [\mathbf{x}_1, ..., \mathbf{x}_M, \mathbf{x}_{M+1}, ..., \mathbf{x}_N]$ where $[\mathbf{x}_1, ..., \mathbf{x}_M]$ and $[\mathbf{x}_{M+1}, ..., \mathbf{x}_N]$ belong to the first and second modality.

### 3.1 The Proposed Invertible Cross-Attention (ICA)

We introduce a novel Invertible Cross-Attention to address the prior limitations in Normalizing Flow-based models. The success of attention mechanisms relies on the capability of exploring the relationship among features via second-order correlations. In particular, the design of the attention layers can be formulated as $\mathrm{Attention}(\mathbf{Q}, \mathbf{K}, \mathbf{V}) = \mathrm{softmax}\left( \frac{\mathbf{Q} \times \mathbf{K}^T}{\sqrt{d}} \right) \mathbf{V}$ where $\mathbf{Q}$, $\mathbf{K}$, and $\mathbf{V}$ are the query, key, and value features obtained by applying linear projection to the input $\mathbf{X}$, and $\times$ is the scale-dot product. The query and key are used to learn the attention weights via a scaled dot product. Then, this attention information is accumulated into the value vector, which allows the final outputs

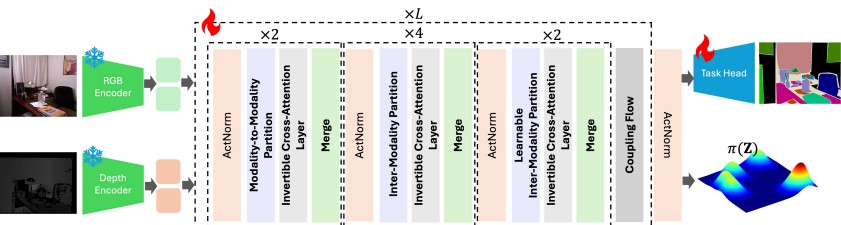

Figure 2: Our Proposed Multimodal Attention-based Normalizing Flows (MANGO) Approach to Fusion Learning.

to carry the correlation among tokens. Inspired by this attention design, we propose the ICA within the coupled layer as in Eqn. (3).

$$\mathbf{X}_1, \mathbf{X}_2 = \text{partition}([\mathbf{x}_1, ..., \mathbf{x}_N])$$
$$\mathbf{Q} = \text{LN}(\text{LP}(\mathbf{X}_1)), \quad \mathbf{K} = \text{LN}(\text{LP}(\mathbf{X}_1)), \quad \mathbf{V} = \mathbf{X}_2$$
$$\mathbf{Y}_1 = \mathbf{X}_1, \quad \mathbf{Y}_2 = \text{softmax}\left(\frac{\mathbf{Q} \times \mathbf{K}^T}{\sqrt{d}}\right)\mathbf{V} \tag{3}$$
$$\mathbf{Y} = \text{merge}([\mathbf{Y}_1, \mathbf{Y}_2])$$

where LN is the layer norm, LP is the linear projection, and $d$ is the feature dimension. This cross-attention mechanism aims to model the inter-token interaction via the attention weights. The attention information in the first patch of inputs ($\mathbf{X}_1$) is embedded into the second patch of inputs ($\mathbf{X}_2$). By scaling into multiple invertible cross-attention layers and alternating the token partitions, our proposed approach can efficiently capture the correlation among inputs, especially in multimodal data, since the attention information across input partitions is exchanged interwisely.

**Invertibilty.** The success of the current state-of-the-art of large-scale generative models, e.g., Large Language Models (LLM) [53], Large Vision-Language Models (LVM) [28, 27], relies on the auto-regressive modeling. Indeed, the auto-regressive form naturally aligns with the nature of the data, where each input token depends on the previous ones. This modeling

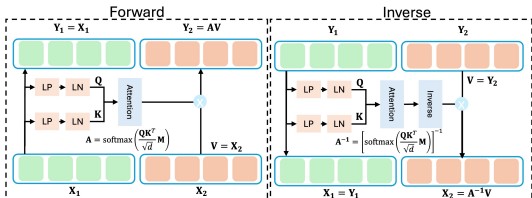

Figure 3: Our Invertible Cross-Attention (ICA).

approach can model the highly complex dependencies within the multimodal data and maintain consistency and coherence. In our learning approach, we propose to model the invertible attention layer via the auto-regressive form. In particular, our ICA layer in Eqn. (3) can be reformed as in Eqn. (4).

$$\mathbf{Y}_2 = \text{softmax}\left(\frac{\mathbf{Q} \times \mathbf{K}^T}{\sqrt{d}}\mathbf{M}\right)\mathbf{V} \tag{4}$$

where $\mathbf{M}$ is the upper triangular matrix to ensure the auto-regressive modeling property. Under this form, the inverse process of our ICA can be formulated as in Eqn. (5).

$$\mathbf{Y}_1, \mathbf{Y}_2 = \text{partition}([\mathbf{y}_1, ..., \mathbf{y}_N])$$
$$\mathbf{Q} = \text{LN}(\text{LP}(\mathbf{Y}_1)), \quad \mathbf{K} = \text{LN}(\text{LP}(\mathbf{Y}_1)), \quad \mathbf{V} = \mathbf{Y}_2$$
$$\mathbf{X}_1 = \mathbf{X}_1, \quad \mathbf{X}_2 = \left[\text{softmax}\left(\frac{\mathbf{Q} \times \mathbf{K}^T}{\sqrt{d}}\mathbf{M}\right)\right]^{-1}\mathbf{V} \tag{5}$$
$$\mathbf{X} = \text{merge}([\mathbf{X}_1, \mathbf{X}_2])$$

Fig. 3 illustrates the forward and inverse process of the ICA layer. Let $\mathbf{A} = \text{softmax}\left(\frac{\mathbf{Q} \times \mathbf{K}^T}{\sqrt{d}}\mathbf{M}\right)$ be the cross-attention matrix. Thanks to the auto-regressive modeling, the inverse matrix of $\mathbf{A}$ always exists since $\mathbf{A}$ is the upper triangular matrix. It should be noted that the diagonal of $\mathbf{A}$ is always greater than 0 due to the softmax properties. Therefore, our approach can efficiently ensure the invertibility of the cross-attention layers. Inspired by [48, 64], $d$ will be a learnable parameter to capture a general scale.

**Tractability.** One of the crucial properties required by the Normalizing Flow-based model is the tractability of the determinant of the Jacobian matrix, i.e., $\det\left(\frac{\partial \mathbf{Y}}{\partial \mathbf{X}}\right)$. Formally, the determinant of the Jacobian matrix of our ICA can be formed as in Eqn. (6).

$$\det\left(\frac{\partial \mathbf{Y}}{\partial \mathbf{X}}\right) = (\det(\mathbf{A}))^{N/2} = \det\left(\text{softmax}\left(\frac{\mathbf{Q} \times \mathbf{K}^T}{\sqrt{d}}\mathbf{M}\right)\right)^{N/2} \tag{6}$$

Since $\mathbf{A}$ is an upper block triangular matrix due to the autoregressive form, the determinant can be simply computed as the product along the diagonal of the matrix.

### 3.2 The Partitioning Approaches to Cross-Modality Attention

As shown in Eqn. (3), the partitioning method plays a vital role in learning the correlation across modalities since it will decide which attention information will be exchanged within the invertible

cross-attention layers. To support the correlation learning across modalities, we propose to design three different partitioning approaches to capture different types of cross-modality attention (Fig. 4).

For simplicity, we rewrite the multimodal input $\mathbf{X} = [\mathbf{x}_1, ..., \mathbf{x}_M, \mathbf{x}_{M+1}, ..., \mathbf{x}_N]$ as $\mathbf{X} = [\mathbf{x}_1^A, ..., \mathbf{x}_M^A, \mathbf{x}_1^B, ..., \mathbf{x}_K^B]$ where $K$ is the number of tokens of the second modality, i.e., $N = M + K$.

**Modality-to-Modality Cross-Attention (MMCA).** To capture the attention from the first to the second modality (or vice versa), the partition function in Eqn. (3) can be formed as:

$$\underbrace{\begin{cases} \mathbf{X}_1 &= [\mathbf{x}_1^A, ..., \mathbf{x}_M^A] \\ \mathbf{X}_2 &= [\mathbf{x}_1^B, ..., \mathbf{x}_K^B] \end{cases}}_{\text{partition}_{A \to B}^{MMCA}} \text{ or } \underbrace{\begin{cases} \mathbf{X}_1 &= [\mathbf{x}_1^B, ..., \mathbf{x}_K^B] \\ \mathbf{X}_2 &= [\mathbf{x}_1^A, ..., \mathbf{x}_M^A] \end{cases}}_{\text{partition}_{B \to A}^{MMCA}} \quad (7)$$

Figure 4: Our Proposed Partitioning Approaches: Modality-to-Modality Cross-Attention (Left). Inter-Modality Cross-Attention (Middle). Learnable Inter-Modality Cross-Attention (Right).

where $\text{partition}_{A \to B}^{MMCA}$ and $\text{partition}_{B \to A}^{MMCA}$ are the Modality-to-Modality partitioning methods.

While the first partition method $\text{partition}_{A \to B}^{MMCA}$ allows the ICA layers to capture the correlation of the first modality to the second modality, $\text{partition}_{B \to A}^{MMCA}$, will model the attention in the reverse direction, i.e., from the second to the first modality. Under this approach, the intra-attention information can be exchanged across modalities effectively. Then, the merging method of the corresponding partitioning function can be formulated as in Eqn. (8).

$$\underbrace{\text{merge}([\mathbf{Y}_1 \mathbf{Y}_2]) = [\mathbf{Y}_1, \mathbf{Y}_2]}_{\text{partition}_{A \to B}^{MMCA}} \text{ or } \underbrace{\text{merge}([\mathbf{Y}_1 \mathbf{Y}_2]) = [\mathbf{Y}_2, \mathbf{Y}_1]}_{\text{partition}_{B \to A}^{MMCA}} \quad (8)$$

This merging method aims to maintain the consistency of the token positions by reorganizing the positions of the output tokens corresponding to their original ones in input $\mathbf{X}$.

**Inter-Modality Cross-Attention (IMCA).** To model the inter-attention across modalities, our partitioning function can be formulated as follows,

$$\text{partition}^{IMCA} = \begin{cases} \mathbf{X}_1 &= [\underbrace{\mathbf{x}_1^A, ..., \mathbf{x}_{M/2}^A}_{\mathbf{x}_A^1}, \underbrace{\mathbf{x}_1^B ... \mathbf{x}_{K/2}^B}_{\mathbf{x}_B^1}] \\ \mathbf{X}_2 &= [\underbrace{\mathbf{x}_{M/2+1}^A, ..., \mathbf{x}_M^A}_{\mathbf{x}_A^2}, \underbrace{\mathbf{x}_{K/2+1}^B ... \mathbf{x}_K^B}_{\mathbf{x}_B^2}] \end{cases} \quad (9)$$

where $\text{partition}^{IMCA}$ is the inter-modality partitioning method. Our partitioning method has four different ways to divide partitions, i.e., $(\mathbf{X}_1, \mathbf{X}_2) \in \{([\mathbf{X}_A^1, \mathbf{X}_B^1], [\mathbf{X}_A^2, \mathbf{X}_B^2]), ([\mathbf{X}_A^1, \mathbf{X}_B^2], [\mathbf{X}_A^2, \mathbf{X}_B^1]), ([\mathbf{X}_A^2, \mathbf{X}_B^1], [\mathbf{X}_A^1, \mathbf{X}_B^2]), ([\mathbf{X}_A^2, \mathbf{X}_B^2], [\mathbf{X}_A^1, \mathbf{X}_B^1])\}$. Our inter-modality partitioning approach allows the cross-attention layer to capture the correlation across modalities efficiently. Then, the inter-modality attention in the first partition $(\mathbf{X}_1)$ can be embedded into the second partition $(\mathbf{X}_2)$. Then, the merging method of the partitioning function $\text{partition}^{IMCA}$ can be formulated as in Eqn. (10).

$$\text{merge}(\mathbf{Y}_1, \mathbf{Y}_2) = [\mathbf{Y}_A^1, \mathbf{Y}_A^2, \mathbf{Y}_B^1, \mathbf{Y}_B^2] \quad (10)$$

where $\mathbf{Y}_A^1, \mathbf{Y}_A^2, \mathbf{Y}_B^1, \mathbf{Y}_B^2$ are the corresponding outputs of $\mathbf{X}_A^1, \mathbf{X}_A^2, \mathbf{X}_B^1, \mathbf{X}_B^2$ produced by ICA.

**Learnable Inter-Modality Cross-Attention (LICA).** To further improve IMCA learning, we introduce a new Learnable Inter-Modality Cross-Attention as follows,

$$\mathbf{X}' = [\mathbf{x}_1', ..., \mathbf{x}_N'] = \mathbf{X}\mathbf{W}_{per} \quad \text{partition}^{LICA} = \begin{cases} \mathbf{X}_1 &= [\mathbf{x}_1', ..., \mathbf{x}_{N/2}'] \\ \mathbf{X}_2 &= [\mathbf{x}_{N/2+1}', ..., \mathbf{x}_N'] \end{cases} \quad (11)$$

where $\mathbf{W}_{per}$ is the learnable permutation matrix.

To maintain the permutation property of the matrix $\mathbf{W}_{per}$, we adopt the LU Decomposition [21] as $\mathbf{W}_{per} = \mathbf{P}\mathbf{L}(\mathbf{U} + \text{diag}(\mathbf{s}))$, where $\mathbf{P}$ is the fixed permutation matrix, $\mathbf{L}$ and $\mathbf{U}$ are the learnable lower and upper triangular matrices with ones and zeros on the diagonal, and $\mathbf{s}$ is the learnable vector. Since $\mathbf{W}_{per}$ is the permutation matrix, the inverse permutation matrix $\mathbf{W}^{-1}$ can be computed and the Jacobian determinant of $\frac{\partial \mathbf{X}'}{\partial \mathbf{X}}$ can be determined via the vector $\mathbf{s}$, i.e., $\log \det \left| \frac{\partial \mathbf{X}'}{\partial \mathbf{X}} \right| = \sum (\log |\mathbf{s}|)$.

Our approach can efficiently capture the underlying cross-attention across input modalities using the proposed LICA approach. Then, the merging method of the learnable partitioning function can be formed via the inverse permutation $\mathbf{W}_{per}^{-1}$ as follows:

$$\text{merge}([\mathbf{Y}_1, \mathbf{Y}_2]) = [\mathbf{Y}_1, \mathbf{Y}_2]\mathbf{W}_{per}^{-1} \quad (12)$$

### 3.3 Multimodal Latent Normalizing Flows

The typical likelihood-based model has two stages. First, the perceptual compression stage focuses on removing high-frequency details while learning little semantic information. Second, the semantic compression stage will learn the semantic and conceptual composition represented in the data [42]. As a result, the second stage plays a more important role since it is an actual generative model that learns the semantic structures and cross-modality correlations represented in the multimodal data. The original data is often represented in high-dimensional space, e.g., high-resolution images or long sequence data. However, the semantic information of the data can be represented in a much lower-dimensional space since the input space has redundant dimensions. Therefore, based on the intrinsic dimensionality of the input data, we aim to find a perceptually equivalent but computationally efficient space for our multimodal normalizing flow-based approach.

Table 1: Comparison of RGB-D Semantic Segmentation Performance on NYUDv2 and SUN RGB-D with Prior Methods. Our metrics include Pixel Accuracy (Pixel Acc.) (%), Mean Accuracy (mAcc.) (%), Mean Intersection over Union (mIoU) (%).

| Method | Inputs | NYUDv2 | | | SUN RGB-D | | |
|---|---|---|---|---|---|---|---|
| | | Pixel Acc. | mAcc. | mIoU | Pixel Acc. | mAcc. | mIoU |
| CNN-based models | | | | | | | |
| FCN-32s [30] | RGB | 60.0 | 42.2 | 29.2 | 68.4 | 41.1 | 29.0 |
| RefineNet [26] | RGB | 74.4 | 59.6 | 47.6 | 81.1 | 57.7 | 47.0 |
| FuseNet [13] | RGB+D | 68.1 | 50.4 | 37.9 | 76.3 | 48.3 | 37.3 |
| SSMA [62] | RGB+D | 75.2 | 60.5 | 48.7 | 81.0 | 58.1 | 45.7 |
| RDFNet [39] | RGB+D | 76.0 | 62.8 | 50.1 | 81.5 | 60.1 | 47.7 |
| AsymFusion [67] | RGB+D | 77.0 | 64.0 | 51.2 | - | - | - |
| CEN [66] | RGB+D | 77.7 | 65.0 | 52.5 | 83.5 | 63.2 | 51.1 |
| Transformer-based models | | | | | | | |
| DPLNet [6] | RGB+D | - | - | 59.3 | - | - | 52.8 |
| DFormer [71] | RGB+D | - | - | 57.2 | - | - | 52.5 |
| EMSANet [44] | RGB+D | - | - | 59.0 | - | - | 50.9 |
| W/O Fusion (Tiny) [65] | RGB | 75.2 | 62.5 | 49.7 | 82.3 | 60.6 | 47.0 |
| Feature Concat (Tiny) [65] | RGB+D | 76.5 | 63.4 | 50.8 | 82.8 | 61.4 | 47.9 |
| TokenFusion (Tiny) [65] | RGB+D | 78.6 | 66.2 | 53.3 | 84.0 | 63.3 | 51.4 |
| W/O fusion (Small) [65] | RGB | 76.0 | 63.0 | 50.6 | 82.9 | 61.3 | 48.1 |
| Feature Concat (Small) [65] | RGB+D | 77.1 | 63.8 | 51.4 | 83.5 | 62.0 | 49.0 |
| TokenFusion (Small) [65] | RGB+D | 79.0 | 66.9 | 54.2 | 84.7 | 64.1 | 53.0 |
| GeminiFusion (MiT-B5) [20] | RGB+D | 80.3 | 70.4 | 57.7 | 83.8 | 65.3 | 53.3 |
| MANGO | RGB+D | **81.5** | **71.6** | **59.2** | **83.9** | **67.2** | **54.1** |

**Perceptual Compression.** Inspired by the success of prior work [42], we propose to project the data into a much lower-dimensional feature space but with more meaningful information in the representation. Let $\mathcal{E}$ be the encoder that maps the input $\mathbf{X}$ in to the latent feature $\mathbf{F}$, i.e., $\mathbf{F} = \mathcal{E}(\mathbf{X})$. Then, the decoder $\mathcal{D}$ will map the features back into its original data space, i.e., $\mathbf{X} = \mathcal{D}(\mathbf{F})$. The design of encoder $\mathcal{E}$ and the decoder $\mathcal{D}$ can be varied, e.g., PCA, Autoencoder [41, 14]. However, to achieve the best capability of perceptual compression, we adopt the autoencoder approach [41, 14] to develop the encoder and the decoder. This approach can provide a new input space perceptually equivalent to the data space while maintaining the lower-dimensional space.

**Latent Normalizing Flow-based Model.** Instead of modeling the multimodal data $\mathbf{X}$ on its original high-dimensional space, we propose to model the data distribution via its multimodal feature $\mathbf{F}$ on the latent space as $p(\mathbf{F}) = \pi(\mathbf{Z}) \left| \frac{\partial G(\mathbf{F})}{\partial \mathbf{F}} \right|$. We named this method the Multimodal Attention-based Normalizing Flow Approach with a Latent Model. With our approach, the flow-based model does not need to learn to perform perceptual compression on high-dimensional data. Instead, our normalizing flow approach will focus on learning the semantic information and correlation of multimodal data. As a result, our model exhibits better scaling properties while using an efficient computational cost. In addition, the bijective network $G$ designed via our Invertible Cross-Attention layers offers better multimodal modeling via second-order correlation learning.

**Attention-based Normalizing Flow Network.** Our bijective network $G$ (Fig. 2) consists of $L$ blocks where each block consists of eight invertible cross-attention layers and a coupling layer [5]. The first two cross-attention layers adopt the MMCA partitioning. The following four cross-attention layers perform different IMCA partitioning approaches. The next two layers utilize LICA Cross-Attention layers. Then, the coupling layer is adopted to increase the inner expressiveness of the bijective blocks.

**Learning MANGO With Task Specific.** Given the multimodal input $\mathbf{X}$ and the label of a specific task $\mathbf{Y}$, MANGO can be jointly optimized via the negative log-likelihood and the task-specific learning objective as in Eqn. (13).

$$\theta^* = \arg\min_{\theta} \mathbb{E}_{\mathbf{X},\mathbf{Y}} \left[ -\left( \log \pi(\mathbf{Z}) + \log \left| \frac{\partial G(\mathbf{F})}{\partial \mathbf{F}} \right| \right) + \mathcal{L}_{task}(\hat{\mathbf{Y}}, \mathbf{Y}) \right] \tag{13}$$

where $\mathbf{Z} = G(\mathcal{E}(\mathbf{X}))$, $\hat{\mathbf{Y}}$ is the prediction of the corresponding task, $\mathcal{L}_{task}$ is the loss of the corresponding prediction task, and $\theta$ is the parameters of the model.

## 4 Experimental Results

### 4.1 Implementation and Benchmarks

**Implementation.** Our bijective network $G$ consists of $L = 12$ cross-attention blocks. For the perceptual compression encoder $\mathcal{E}$, we adopt the visual encoder of [14] for both RGB and Depth

images. We utilize the text encoder from [40] for the textual data. For fair comparisons, we use the task heads of semantic segmentation, image translation, and movie genre classification from [70, 65]. Our experiments are conducted on the 4 NVIDIA A100 GPUS. Our training uses the same learning hyper-parameters from [65] and an input image size of $256 \times 256$ for fair comparisons.

**Semantic Segmentation.** This task uses the two homogeneous inputs of RGB and Depth images to predict the segmentation maps. We perform experiments on NYUDv2 [36] and SUN RGB-D [45]. While NYUDv2 consists of 795/654 images for training and testing splits, SUN RGB-D includes 5,285/5,050 samples for training and testing.

**Image-to-Image Translation.** Following the standard protocol in [65], we adopt the Taskonomy [75] for the multimodal image translation task. This large-scale indoor scene dataset provides over ten multimodal, e.g., RGB, Depth, Normal, Shade, Texture, Edge, etc. We use a subset of 1,000 high-quality images for training and 500 for validation.

**MM-IMDB Movie Genre Classification.** MM-IMDB is a large-scale multimodal dataset for movie genre classification. We adopt the training and testing split of [70] for fair comparisons. In particular, the data in our experiments consists of 15,552 data for training and 2,608 for validation. In this multimodal learning task, we use the inputs from two modalities of images and texts.

## 4.2 Comparison with State-of-the-Art Methods

**Semantic Segmentation.** Table 1 presents our results compared to prior multimodal methods on multimodal semantic segmentation. Our results show the proposed approach achieves state-of-the-art performance on both the NYUDv2 and SUN RGB-D datasets. Our model consistently outperforms the prior methods in all evaluation metrics and datasets. In particular, the mIoU results of our proposed approach are higher than GeminiFusion by 1.5% and 0.6% on the two datasets. Our results have illustrated that our explicit modeling of multimodal fusion has shown a clear advantage over the prior fusion methods [20, 65]. Fig. 5 visualizes the results of our fusion approach via our normalizing flows compared to the prior fusion method, i.e., TokenFusion [65].

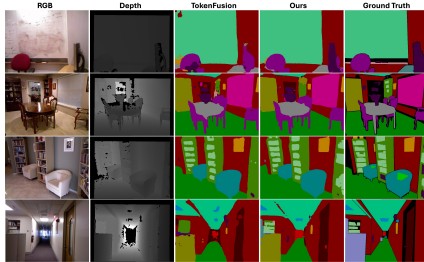

Figure 5: Qualitative Comparison on NYUDv2 Benchmark.

**Image-to-Image Translation.** We present our results on the five different learning settings of multimodal Image-to-Image Translation as shown in Table 2. Our results consistently outperform prior methods in five different multimodal learning settings. In particular, compared to prior GeminiFusion [20], our models have gained better FID scores, i.e., lower than GeminiFusion by 1.71 and 29.37 on benchmarks of Shade+Texture → RGB and Depth+Normal → RGB. These results further confirm the outstanding capability of our approach to capture complex correlations across modalities.

Table 2: Comparison of Multimodal Image Translation Performance on Taskonomy with Prior Multimodal Methods. We use evaluations of FID/KID ($\times 10^{-2}$) for the RGB target and MAE ($\times 10^{-1}$)/MSE ($\times 10^{-1}$) for Normal, Shade, and Depth targets.

| Method | Shade+Texture →RGB (↓) | Depth+Normal →RGB (↓) | RGB+Shade →Normal (↓) | RGB+Normal →Shade (↓) | RGB+Edge →Depth (↓) |
|---|---|---|---|---|---|
| CNN-based models | | | | | |
| Concat [66] | 78.82/3.13 | 99.08/4.28 | 1.34/2.85 | 1.28/2.02 | 0.33/0.75 |
| Self-Attention [63] | 73.87/2.46 | 96.73/3.95 | 1.26/2.76 | 1.18/1.76 | 0.30/0.70 |
| Align. [46] | 92.30/4.20 | 105.03/4.91 | 1.52/3.25 | 1.41/2.21 | 0.45/0.90 |
| CEN [66] | 62.63/1.65 | 84.33/2.70 | 1.12/2.51 | 1.10/1.72 | 0.28/0.66 |
| Transformer-based models | | | | | |
| Feature Concat (Tiny) [65] | 76.13/2.85 | 102.70/4.74 | 1.52/3.15 | 1.33/2.20 | 0.40/0.83 |
| TokenFusion (Tiny) [65] | 50.40/1.03 | 76.35/2.19 | 0.73/1.83 | 0.95/1.54 | 0.21/0.57 |
| Feature Concat (Small) [65] | 72.55/2.39 | 96.04/4.09 | 1.18/2.73 | 1.30/2.07 | 0.35/0.68 |
| TokenFusion (Small) [65] | 43.92/0.94 | 70.13/1.92 | 0.58/1.51 | 0.79/1.33 | 0.16/0.47 |
| GeminiFusion [20] | 41.32/0.81 | 96.98/3.71 | 0.65/1.69 | - | 0.20/0.49 |
| MANGO | **39.61/0.77** | **67.61/1.54** | **0.52/1.12** | **0.62/0.96** | **0.17/0.33** |

**MM-IMDB Movie Genre Classification.** Table 3 presents the results of our approach on the multimodal classification benchmarks. As shown in Table 3, our proposed approach outperforms the prior methods on both micro-average and macro-average F1 scores and achieves state-of-the-art performance. Particularly, our results of Micro and Macro F1 scores remain higher than the prior method [69] by 3.5% and 4.9%. These results illustrate that our approach performs better on homogeneous and heterogeneous inputs. Fig. 6 visualizes our results on the Image-to-Image translation benchmark.

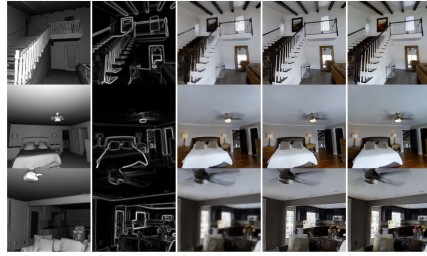

Figure 6: Qualitative Comparison on Image-to-Image Benchmark.

## 4.3 Ablation Studies

**Effectiveness of Invertible Cross-Attention Layers.** To illustrate the impact of our proposed invertible cross-attention layers, we conduct experiments to compare our proposed layers with other flow models, i.e., Affine Coupling Layer [5], Glow [21], Flow++ [15], and AttnFlow [48]. As shown in Table 4, our proposed invertible cross-attention layer consistently outperforms the prior coupling methods. In particular, the mIoU results of our method achieved up to 59.2% and 54.1% on both NYUDv2 and SUN RGBD benchmarks. These results clearly illustrate the advantages of our proposed method for modeling correlations and complex structures in multimodal data.

**Effectiveness of Different Partitioning Approach.** Table 5 presents the experimental results of different partitioning approaches. As shown in the results, using Modality-to-Modality and Inter-Modality Cross-attention, the mIoU results on both NYUDv2 and SUN RGBD benchmarks have achieved 58.0% and 53.7%. Moreover, when the Learnable Inter-Modality Cross-Attention is adopted, our mIoU results are further improved by 59.2% and 54.1% compared to those without LICA. The experimental results have confirmed the effectiveness of our proposed approach in modeling correlation across modalities via our cross-attention mechanism.

**Effectiveness of Latent Model.** These experiments study the effectiveness of our latent model approach. As shown in Table 6, the performance of our multimodal normalizing flow-based models is consistently improved on both semantic segmentation benchmarks using the latent model. The proposed method achieves the SoTA results where the mIoU results of our best model have achieved 59.2% and 54.1% on NYUDv2 and SUN RGBD benchmarks. The results have highlighted the advantages of using the perceptual compression encoder to produce a lower but more efficient representation space.

**Effectiveness of Number of Cross-Attention Blocks.** Table 6 illustrates the results of our approach using different numbers ($L$) of cross-attention blocks. As in our results, the performance of multimodal segmentation models using a deeper network results in better performance. In particular, using $L = 12$ blocks of invertible cross-attention blocks, the mIoU performance on NYUDv2 and SUN RGBD benchmarks has reached up to 59.2% and 54.1%, respectively. While fewer blocks may result in lower computational costs, the deeper model can exploit better correlation of features in multimodal data.

Table 3: Comparison of Movie Genre Classification Performance on the MM-IMDB dataset with Prior Multimodal Methods. Our metrics include the Micro-Average and Macro-Average F1 Scores.

| Method | Modality | Micro F1 (%) | Macro F1 (%) |
|---|---|---|---|
| Image Network [70] | I | 40.0 | 25.3 |
| Text Network [70] | T | 59.2 | 47.2 |
| Late Fusion [24] | I+T | 59.6 | 51.0 |
| LRTF [29] | I+T | 59.2 | 49.3 |
| MI-Matrix [19] | I+T | 58.5 | 48.4 |
| DynMM [70] | I+T | 60.4 | 51.6 |
| COCA [74] | I+T | 67.7 | 62.6 |
| MFM [3] | I+T | 67.5 | 61.6 |
| BLIP [22] | I+T | 67.4 | 62.8 |
| ReFNet [43] | I+T | 68.0 | 58.7 |
| BridgeTow [69] | I+T | 68.2 | 63.3 |
| MANGO | I+T | **71.7** | **68.2** |

Table 4: Effectiveness of Invertible Layers.

| Layer | NYUDv2 | | | SUN RGBD | | |
|---|---|---|---|---|---|---|
| | Pixlel Acc. | mAcc. | mIoU | Pixlel Acc. | mAcc. | mIoU |
| Coupling Layer [5] | 76.0 | 63.4 | 50.8 | 79.8 | 59.9 | 48.5 |
| Glow [21] | 77.0 | 66.4 | 53.0 | 80.3 | 61.9 | 49.1 |
| Flow++ [15] | 77.5 | 68.1 | 54.2 | 81.5 | 62.0 | 50.5 |
| AttnFlow [48] | 79.5 | 69.9 | 56.5 | 82.5 | 65.1 | 52.2 |
| MANGO | **81.5** | **71.6** | **59.2** | **83.9** | **67.2** | **54.1** |

Table 5: Effectiveness of Partitioning Approaches.

| MMCA | IMCA | LICA | NYUDv2 | | | SUN RGBD | | |
|---|---|---|---|---|---|---|---|---|
| | | | Pixlel Acc. | mAcc. | mIoU | Pixlel Acc. | mAcc. | mIoU |
| ✓ | | | 79.3 | 68.8 | 56.4 | 82.4 | 64.6 | 51.3 |
| ✓ | ✓ | | 80.2 | 70.8 | 58.0 | 83.3 | 66.2 | 53.7 |
| ✓ | ✓ | ✓ | **81.5** | **71.6** | **59.2** | **83.9** | **67.2** | **54.1** |

## 5 Conclusions

Our paper has introduced a new explicit modeling approach to multimodal fusion learning via the Attention-based Normalizing Flow-Based Model. Our proposed ICA layers with three different cross-attention mechanisms have efficiently captured the complex structure and underlying correlations in multimodal data. We have also introduced a new latent approach to normalizing flows to increase our scalability to multimodal data. Our intensive experiments on three standard benchmarks, i.e., Semantic Segmentation, Image-to-Image Translation, and Movie Genre Classification, have shown the effectiveness of our approach. Our study has demonstrated the effectiveness of invertible cross-attention layers in multimodal learning under selected hyperparameters and benchmarks. However, it still remains limitations in objective balancing and scalability. The detailed limitations are discussed in our appendix.

Table 6: Effectiveness of Latent Model.

| # Blocks | Latent Model | NYUDv2 | | | SUN RGBD | | |
|---|---|---|---|---|---|---|---|
| | | Pixlel Acc. | mAcc. | mIoU | Pixlel Acc. | mAcc. | mIoU |
| 6 | ✗ | 75.9 | 63.5 | 51.0 | 79.4 | 59.1 | 47.3 |
| | ✓ | 77.5 | 65.8 | 52.3 | 79.6 | 60.5 | 48.1 |
| 8 | ✗ | 78.0 | 65.5 | 53.1 | 80.8 | 60.8 | 49.4 |
| | ✓ | 78.1 | 65.3 | 54.1 | 84.4 | 60.0 | 51.4 |
| 12 | ✗ | 80.7 | 70.4 | 58.0 | 83.4 | 65.8 | 53.5 |
| | ✓ | **81.5** | **71.6** | **59.2** | **83.9** | **67.2** | **54.1** |

**Acknowledgment.** This work is partly supported by NSF CAREER (No. 2442295), NSF SCH (No. 2501021), NSF E-RISE (No. 2445877), NSF SBIR Phase 2 (No. 2247237) and USDA/NIFA Award. We also acknowledge the Arkansas High-Performance Computing Center (HPC) for GPU servers. Nitin Agarwal's participation was supported by U.S. NSF (OIA-1946391, OIA-1920920), AFOSR (FA9550-22-1-0332), ARO (W911NF-23-1-0011, W911NF-24-1-0078, W911NF-25-1-0147), ONR (N00014-21-1-2121, N00014-21-1-2765, N00014-22-1-2318), AFRL, DARPA, Australian DSTO Strategic Policy Grants Program, Arkansas Research Alliance, the Jerry L. Maulden/Entergy Endowment, and the Donaghey Foundation at the University of Arkansas at Little Rock.

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

# Appendices

## A  Additional Ablation Studies

**Effectiveness of Number of Cross-Attention Blocks.** We conducted an ablation study with 16 cross-attention blocks. As shown in Table 7, although using more cross-attention blocks will increase the computation, it helps to enhance the model performance.

Table 7: Effectiveness of Number of Cross-Attention Blocks.

| # Blocks | NYUv2 | | | SUN RGBD | | |
|---|---|---|---|---|---|---|
| | Acc. | mAcc. | mIoU | Acc. | mAcc. | mIoU |
| 6 | 77.5 | 65.8 | 52.3 | 79.6 | 60.5 | 48.1 |
| 8 | 78.1 | 65.3 | 54.1 | 84.4 | 60.0 | 51.4 |
| 12 | 81.5 | 71.6 | 59.2 | 83.9 | 67.2 | 54.1 |
| 16 | 83.1 | 75.1 | 61.7 | 85.4 | 68.7 | 55.6 |

**Computational Cost.** As shown in Table 8, the parameters, GFLOPs, and inference time of our method are competitive with prior methods. Meanwhile, we achieved state-of-the-art performance on two segmentation benchmarks.

Table 8: The Comparision of Computational Cost.

| Method | NYUDv2 mIOU | SUN RGB-D mIOU | PARAMS | GFLOPS | Inference Time |
|---|---|---|---|---|---|
| TokenFusion [65] | 54.2 | 53.0 | 45.9M | 108 | 126 ms |
| GeminiFusion [20] | 57.7 | 53.3 | 75.8M | 174 | 153 ms |
| MANGO | 59.2 | 54.1 | 72.9M | 152 | 144 ms |

**Attention Visualization.** As shown in Figure 7, our Invertible Cross-Attention layer can capture the attention interaction from the region in the depth image (red box) to the RGB image. This result has illustrated the effectiveness of our proposed attention layer in capturing the correlation across modalities.

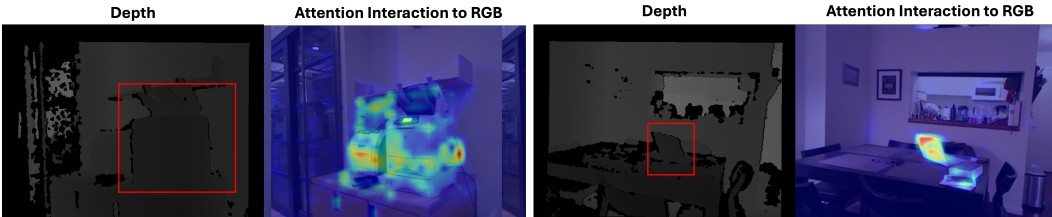

Figure 7: The Attention Visualization of ICA Layer.

## B  Dicussion of Limitations

Our experiments have chosen a set of learning hyper-parameters and benchmarks to support our hypothesis. However, our work could contain several limitations. Our work studied the effectiveness of our proposed invertible cross-attention layers in multimodal learning. Thus, the investigation of balance weights among learning objectives has not been fully exploited, and we leave this experiment as our future work. Due to computation limitations, our experiments are limited to the standard scale of the benchmarks. However, we hypothesize that the proposed approaches can generalize to larger-scale data and benchmark settings according to the fundamental theories presented in our paper.

