# OpenReview forum: "MANGO: Multimodal Attention-based Normalizing Flow Approach to Fusion Learning"
_NeurIPS.cc/2025/Conference — NeurIPS 2025 poster_

### Official Review · Reviewer_XZY1 · 2025-06-24

**Clarity:** 2
**Significance:** 2
**Originality:** 3
**Rating:** 4
**Confidence:** 4

**Summary:**

This paper identifies a key limitation in existing multimodal fusion approaches: leveraging Transformer-based attention mechanisms for fusion hinders the model's ability to effectively capture the complex structures and correlations inherent in multimodal data. To address this issue, the authors propose a method called Multimodal Attention-based Normalizing Flow (MANGO), which builds a multimodal modeling framework using Invertible Cross-Attention (ICA) layers. By introducing three types of cross-attention mechanisms, MANGO effectively captures the intricate structures and relationships within multimodal inputs. Experiments conducted on three different multimodal tasks validate the effectiveness of the proposed method.

**Questions:**

N/A

**Ethical Concerns:**

["NO or VERY MINOR ethics concerns only"]

**Final Justification:**

The authors have well responsed the first two problems.

For the third problem, the author responded "In the MM-IMDB classification task, our method outperforms strong baselines, which suggests that MANGO can effectively leverage both modalities without overfitting the stronger modality." However, in multimodal tasks, determining whether this issue has been resolved should NOT solely rely on the final performance, as this could be due to the performance boost from better leveraging the stronger modality. We believe the verification should proceed as follows: Since the author claims that existing methods have the above issue, the first step should be to present the accuracy of each modality in a multimodal model trained with the fusion approach of existing methods. Then, the accuracy of each modality in the multimodal model trained with the proposed method should be shown. If the accuracy of the weaker modality improves while the accuracy of the stronger modality does not decrease, it would strongly demonstrate the effectiveness of the proposed method.

**Limitations:**

Yes

**Quality:**

2

**Strengths And Weaknesses:**

Strengths：

1）The methodology and evaluation metrics used in the paper are appropriate.

2）The authors propose a multimodal fusion method from the perspective of explicit modeling, which is relatively novel.

Weaknesses：

1）Although the authors point out in the Introduction that "implicit modeling fails to fully capture the alignment and structural relationships across modalities," the discussion on the underlying causes and theoretical foundations of this phenomenon is relatively weak. The paper leans more toward phenomenological description, lacking rigorous mathematical definitions and theoretical explanations, which makes this concept feel more like an intuition rather than a well-grounded theoretical proposition.

2）In Section 4.2 "Semantic Segmentation," the authors claim that “Our model consistently outperforms the prior methods in all evaluation metrics and datasets.” However, upon reviewing the experimental results in Table 1, it is evident that the performance on the Pixel Accuracy metric for the SUN RGB-D dataset is not the best. Moreover, the visualizations in Figure 5 do not clearly demonstrate the advantages of the proposed method.

3）In the Introduction, the authors mention that existing implicit multimodal fusion methods tend to overly rely on a single modality when it provides more data or stronger signals, leading to suboptimal performance. However, the paper does not include any experiments to verify whether MANGO effectively addresses this issue.

---

> ### Author Rebuttal · Authors · 2025-07-30
>
> Dear Reviewer XZY1,
>
> We would like to express our gratitude for your careful reading and valuable feedback. We are glad you encourage that **our multimodal fusion method from the perspective of explicit modeling is novel**. We appreciate your constructive comments and would like to address these points as follows.
>
>
>
>
> [Q1] **Theoretical foundations of limitations of prior implicit modeling**
>
>
> [A1] The core limitation of Transformer-based fusion models lies in their implicit modeling nature, where attention mechanisms operate without a formal probabilistic framework. These models do not explicitly model the joint distribution of multimodal inputs; instead, they focus on contextual alignment, often overlooking subtle or global inter-modal dependencies and failing to preserve modality-specific information. In contrast, our method is grounded in the theory of normalizing flows, which explicitly models the joint distribution of multimodal inputs through invertible, bijective transformations with tractable likelihood computation. Our ICA layer is a mathematically defined transformation whose Jacobian determinant (Eqn. 6) directly contributes to the overall data likelihood (Eqn. 2), enabling exact and interpretable modeling of multimodal structure while guaranteeing lossless information propagation. Moreover, ICA’s autoregressive formulation introduces directed attention flows, allowing the model to efficiently capture cross-modal dependencies. These properties form a rigorous theoretical foundation for MANGO’s ability to model complex multimodal correlations, which are consistently validated through our experiments and ablation studies.
>
>
>
>
> [Q2] **Performance on the Pixel Accuracy metric for the SUN RGB-D dataset and Visualization of Figure 5**
>
>
> [A2] While it is true that our method does not achieve the highest score on Pixel Accuracy for the SUN RGB-D dataset, we would like to clarify that Mean Intersection-over-Union (mIoU) is the primary and widely accepted metric for evaluating semantic segmentation performance. Unlike pixel accuracy, which can be biased toward large or dominant classes and may overestimate performance due to background or majority-class pixels, mIoU provides a class-balanced evaluation by measuring the overlap between predicted and ground truth regions across all classes. This makes it a more reliable indicator of true segmentation quality, particularly in multimodal settings with varied object distributions. Our approach consistently achieves the highest mIoU across all datasets, which reflects its effectiveness in modeling cross-modal dependencies.
>
>
> Regarding the visualizations in Figure 5, we appreciate the reviewer’s feedback and acknowledge that the differences may not be immediately obvious due to the challenging nature of the NYUDv2 benchmarks, which contain complex indoor scenes, frequent occlusions, and subtle semantic boundaries. In the final version, we will enhance Figure 5 by adding highlight boxes to draw attention to regions where our method corrects boundary errors or recovers finer details, thereby making the qualitative advantages more interpretable.
>
>
>
>
> [Q3] **Existing implicit multimodal fusion methods tend to overly rely on a single modality**
>
>
> [A3] This challenge has been reflected in our experimental setup. In particular, the MM-IMDB classification task involves highly imbalanced modality signals, where text typically carries stronger semantic information than images. Indeed, as shown in the first two rows of Table 3, using text only yields a significantly higher macro-F1 score (47.2%) than using image only (25.3%), confirming that text carries a much stronger semantic signal in this benchmark. In this benchmark, our method outperforms strong baselines, i.e., BridgeTower and BLIP, by +3.5% micro-F1 and +4.9% macro-F1, indicating that MANGO is able to leverage both modalities effectively without overfitting to the stronger one. This suggests that our explicit modeling of the joint distribution via invertible attention within normalizing flows mitigates the issue of modality dominance by preserving complementary signals.

---

> ### Comment · Reviewer_XZY1 · 2025-08-04
>
> Thank the  authors' efforts for addressing my concerns.
>
> The authors have well responsed the first two problems.
>
> For the third problem,  the author responded  "In the MM-IMDB classification task, our method outperforms strong baselines, which suggests that MANGO can effectively leverage both modalities without overfitting the stronger modality." However, in multimodal tasks, determining whether this issue has been resolved should **NOT** solely rely on the final performance, as this could be due to the performance boost from better leveraging the stronger modality. We believe the verification should proceed as follows: Since the author claims that existing methods have the above issue, the first step should be to present the accuracy of each modality in a multimodal model trained with the fusion approach of existing methods. Then, the accuracy of each modality in the multimodal model trained with the proposed method should be shown. If the accuracy of the weaker modality improves while the accuracy of the stronger modality does not decrease, it would strongly demonstrate the effectiveness of the proposed method.
>
> A small suggestion:  In the related work, "Another approach adopted the neural architecture search to search for appropriate networks for multimodal fusion [63,10]". I notice that the earlier work is EDF [1] than [63,10], so I suggest that the authors accurately cite it.
>
> [1] Evolutionary deep fusion method and its application in chemical structure recognition. IEEE Transactions on Evolutionary Computation, 2021,25(5):883-893 .
>
> Based on the consideration, I recommend  borderline accept.

---

> > ### Author Response · Authors · 2025-08-04
> >
> > Dear Reviewer XZY1,
> >
> > We would like to thank the reviewer for your thoughtful feedback and for raising the rating to a borderline accept. We are encouraged by your recognition of our efforts in the rebuttal and greatly appreciate the constructive comments. Your insights and feedback have helped us to improve our paper, and we will incorporate the suggested reference into our literature review.
> >
> > Thank you very much,
> >
> > Authors

---

### Official Review · Reviewer_eadU · 2025-06-30

**Clarity:** 3
**Significance:** 2
**Originality:** 2
**Rating:** 4
**Confidence:** 4

**Summary:**

The paper proposes a multimodal learning method using normalizing flows. Three fusion approaches are proposed to control the degree of modality mixing, which are placed in the model to progressively increase the mixing degree. Three applications are tested: semantic segmentation, image-to-image translation, and movie genre classification, and superiority in performance is shown.

**Questions:**

Please see the weaknesses.

**Ethical Concerns:**

["NO or VERY MINOR ethics concerns only"]

**Final Justification:**

The authors' rebuttal is appreciated. Some of my previous concerns are cleared, but some are still not.

I more or less agree on the conceptual claim that the invertibility can maintain important inter-modal correlation, but still have difficulty in understanding how such a structure actually leads to improved task performance in various tasks. It would be better to provide some other evidence or deeper analysis (other than the final task performance) that can help readers understand more about whether/how the conceptual advantages of the proposed method actually work, such as what the model actually learned, how the model encoded long-range dependency, what kind important information is kept through the invertibility whereas conventional methods lose, how adaptively the model learns different types of inter-modal interactions appearing in different tasks, etc.
(I guess the additional question by reviewer XZY1 is also in a similar context.)

Since some concerns are cleared, I raise my rating to borderline accept.

**Limitations:**

yes

**Paper Formatting Concerns:**

n.a.

**Quality:**

2

**Strengths And Weaknesses:**

* Strengths

1. The writing is mostly fine.
2. It seems that there are not many prior methods using normalizing flows for multimodal learning.

* Weaknesses

1. It is interesting to use invertible operations for multimodal learning, but it is not clear why invertibility is required/beneficial for multimodal learning. It seems the reverse direction is not used at all anyway.

2. It is said that the existing approaches lack explicit modeling of inter-modal correlation. However, the proposed method relies on ICA, which is similar to the popular cross-modal attention except for the assignment of each modality to query, key, and value. So the modeling capability of the proposed method doesn't seem to go beyond the popular cross-modal attention. It is not clearly explained that except for the assignment, how the proposed method is distinguished, and which part of the proposed method allows explicit modeling unlike existing methods.

3. It is expected to justify better the motivation of the operation in the proposed ICA. Basically, the attention scores are calculated from one modality (green in Fig. 3) and are applied to combine tokens of the other modality (orange in Fig. 3). How can this resolve the research question (i.e., how to explicit model correlation better than existing methods)?

4. It seems that the authors intend to emphasize the universality of the proposed method through multiple applications. While the experimental results show better performance of the proposed method, it is not quite clear why the proposed method works well in different applications. For instance, semantic segmentation requires modeling interaction between depth and RGB, whose embeddings would have high spatial correlation, whereas in the movie genre classification, a pair of image and text would have almost no spatial relationship but more complicated semantic relationship. It would be better to both conceptually explain and analyze learned models/representations to understand how the proposed method allows flexible modeling of different inter-modal relationship.

5. Discussion on how the proposed method is distinguished from existing methods is rather insufficient. In the introduction, the limitations of existing approaches are mentioned only briefly and roughly. The argument at the end of Sec. 2.1 about the limitations of existing approaches (unable to model long-range dependency and complex cross-modal correlation) should be supported better by explicitly demonstrating whether the proposed method can overcome those limitations and how. It is not quite clear whether the three applications of the experiments actually require long-range dependency and complex cross-modal correlation and whether the proposed method actually improves the performance specifically due to better modeling of long-range dependency and complex cross-modal correlation.

6. In the three applications, further theoretical/experimental analysis on how the proposed method is distinguished from the compared methods. For instance, how different the proposed method is from the fusion approaches in GeminiFusion or BridgeTower.

7. Overall, the novelty of the proposed method is rather unclear to me.

---

> ### Author Rebuttal · Authors · 2025-07-30
>
> Dear Reviewer eadU,
>
> We greatly appreciate your insightful review and valuable feedback. We are very happy you think **our writing is good, and our normalizing flows for multimodal learning are distinct from prior methods**. We appreciate your constructive comments and would like to address these points as follows.
>
> [Q1] **Benefits of invertibility in multimodal learning**
>
> [A1] The use of invertible operations in MANGO is motivated by both theoretical and practical advantages in multimodal learning. Unlike prior Transformer-based models that rely on implicit attention mechanisms, our approach explicitly models the joint distribution of multimodal inputs via Normalizing Flows with Invertible Cross-Attention (ICA). This design enables MANGO to preserve modality-specific information and capture complex cross-modal dependencies in a mathematically principled and lossless manner. In contrast, Transformers are not invertible, which can lead to information loss, particularly in fine-grained features or subtle cross-modal correlations during feature aggregation. The invertibility of our proposed approach ensures a bijective, lossless mapping between the input and latent spaces, guaranteeing that no information is discarded. This is an important property when fusing heterogeneous modalities with varying signal strength, as it prevents dominant modalities from overwhelming weaker ones during fusion. Moreover, invertibility enables exact likelihood estimation, providing a tractable and interpretable training objective that standard fusion models often lack. As in Section 4, our proposed approach consistently outperforms Transformer-based fusion models across tasks, validating the effectiveness of our invertible formulation.
>
> [Q2] **How the proposed method is distinguished and allows explicit modeling unlike existing methods.**
>
> [A2] The role and functionality of our ICA layers within our framework are fundamentally different from prior methods. In standard Transformers, attention is used within an implicit modeling paradigm, where cross-modal correlations are learned in a latent space without an explicit probabilistic structure. In contrast, our method embeds ICA within a normalizing flow, which explicitly models the joint probability distribution of multimodal inputs (Eqn. 2). In our approach, ICA is not merely used to refine features, but to construct a bijective, invertible transformation that aligns with the theoretical foundations of normalizing flows. Through this formulation, each ICA layer contributes directly to the likelihood computation (Eqn. 6), enabling tractable, mathematically grounded modeling of inter-modal dependencies. Furthermore, the invertibility of ICA guarantees lossless transformation, allowing us to retain and analyze modality-specific contributions, which is not possible in standard attention modules. The ability to compute exact log-likelihoods and preserve modality-wise information flow is what makes our approach explicit, interpretable, and principled, distinguishing it from prior attention-based fusion methods.
>
> [Q3] **Motivation of the operation in the proposed ICA**
>
> [A3] Our motivation lies in the need for explicit, lossless modeling of cross-modal correlations, which existing Transformer-based fusion methods fail to offer. In prior Transformer-based methods, attention mechanisms operate implicitly, without modeling the joint probability distribution of the modalities. They tend to focus on contextual alignment, often overlooking subtle or global inter-modal dependencies, and do not explicitly preserve modality-specific information. Additionally, Transformer attention is not invertible, which can lead to information loss, especially of fine-grained correlations during feature fusion. In contrast, our ICA is designed within a normalizing flow approach, where each layer contributes to a bijective, mathematically grounded transformation. The directional design of ICA (e.g., attention scores computed from modality A and applied to modality B) allows us to explicitly model how one modality modulates the representation of the other, enabling principled and directional interaction that reflects true conditional dependencies. This operation is key to modeling the joint distribution of multimodal inputs, as shown in Eqns. (2)–(6), and is crucial for computing exact log-likelihoods, which the traditional attention mechanisms cannot achieve. Moreover, our approach introduces structured partitioning (MMCA, IMCA, LICA) to ensure that attention is not only directional but also aligned with specific modeling intents (modality-to-modality, inter-modality, or learnable interactions). The invertibility of ICA ensures lossless mapping between modalities and latent space, preserving all correlation structures. As in Section 4, our design yields consistent performance improvements across tasks, outperforming Transformer-based baselines.
>
> [Q4] **Universality of the proposed method through multiple applications**
>
> [A4] One of our goals is to demonstrate the universality and flexibility of MANGO across diverse multimodal tasks, each with distinct types of inter-modal relationships. The core strength of MANGO lies in its ability to explicitly model joint distributions between modalities via a tractable, invertible architecture. Unlike prior methods that use fixed attention mechanisms across all tasks, our design leverages (1) ICA layers, which allow the model to compute directional cross-modal influences in a lossless, learnable manner, and (2) adaptive partitioning strategies (MMCA, IMCA, LICA), which enable flexible structuring of attention flow depending on the modality pair. For example, in semantic segmentation, the spatial correlation between RGB and depth is naturally exploited via MMCA and IMCA; in contrast, for movie genre classification, where image and text have weaker spatial but richer semantic alignment, LICA enables the model to learn an optimal token permutation for aligning semantic patterns. Our multimodal latent normalizing flow design further supports this flexibility by operating in a compressed semantic space, where modality-specific and cross-modal features are jointly modeled. As shown in Tables 1–3, MANGO consistently improves performance across tasks with homogeneous and heterogeneous modality structures.
>
> [Q5] **Discussion on how the proposed method is distinguished from existing methods is rather insufficient.**
>
> [A5] Existing Transformer-based fusion models rely on implicit attention mechanisms that do not explicitly model joint distributions, and often struggle to capture long-range dependencies or complex inter-modal correlations, especially in settings with unbalanced modality strengths or limited supervision. Our method addresses these limitations by introducing normalizing flows with ICA. Specifically, the ICA layers use autoregressive attention matrices (Eqns. 4–6) to model cross-modal influences in a directional, lossless, and long-range manner. They are embedded in a bijective flow architecture that enables tractable modeling of the joint probability distribution across modalities. This design allows our model to capture both global and local dependencies in a mathematically grounded way. Regarding the experimental tasks: semantic segmentation benefits from long-range spatial dependencies between RGB and depth; image translation requires global cross-modal correlation to reconstruct coherent targets from complementary modalities; and movie genre classification depends on aligning complex semantic signals from visual and textual content. As shown in our experiments and ablations (Tables 1-6), our model consistently outperforms flow-based and Transformer-based baselines.
>
> [Q6] **Further theoretical/experimental analysis on how the proposed method is distinguished from the compared methods**
>
> [A6] Our approach is fundamentally different in both architectural design and modeling principles compared to GeminiFusion and BridgeTower. In particular, GeminiFusion performs early fusion and relies on self-attention layers to implicitly learn cross-modal correlations. Meanwhile, BridgeTower integrates vision and language encoders through interleaved attention blocks without modeling the joint distribution explicitly. In contrast, MANGO adopts an explicit modeling paradigm by using normalizing flows to estimate the joint probability distribution of multimodal inputs. Our Invertible Cross-Attention (ICA) layers contribute directly to tractable likelihood computation (Eqns. 2–6), and our partitioning strategies (MMCA, IMCA, LICA) offer structured, interpretable modeling of modality-specific and inter-modal dependencies. As shown in Tables 1-3, MANGO consistently outperforms GeminiFusion and BridgeTower by a large margin. Our results have validated that our explicit, invertible modeling framework brings measurable advantages in real-world multimodal tasks.
>
> [Q7] **The novelty of the proposed method is rather unclear to me.**
>
> [A7] We would like to emphasize that our method introduces a new class of explicit multimodal fusion models that combine normalizing flows with invertible attention mechanisms, which, to our knowledge, has not been explored in prior work. Unlike standard Transformer-based approaches (e.g., GeminiFusion, BridgeTower) that rely on implicit attention fusion, our method embeds a novel Invertible Cross-Attention (ICA) mechanism within a tractable normalizing flow framework. ICA is not only attention-based but also invertible and autoregressive, enabling exact joint likelihood modeling and lossless information preservation, a key theoretical and practical distinction. Our contributions are further supported by Reviewer dFiC, who acknowledges our approach as “novel ... addressing key challenges in cross-modality correlation”, and Reviewer XZY1, who agrees that “the proposed method from the perspective of explicit modeling is relatively novel.”

---

> > ### Comment · Reviewer_eadU · 2025-08-05
> >
> > The authors' rebuttal is appreciated. Some of my previous concerns are cleared, but some are still not.
> >
> > I more or less agree on the conceptual claim that the invertibility can maintain important inter-modal correlation, but still have difficulty in understanding how such a structure actually leads to improved task performance in various tasks. It would be better to provide some other evidence or deeper analysis (other than the final task performance) that can help readers understand more about whether/how the conceptual advantages of the proposed method actually work, such as what the model actually learned, how the model encoded long-range dependency, what kind important information is kept through the invertibility whereas conventional methods lose, how adaptively the model learns different types of inter-modal interactions appearing in different tasks, etc.
> > (I guess the additional question by reviewer XZY1 is also in a similar context.)
> >
> > Since some concerns are cleared, I raise my rating to borderline accept.

---

> > > ### Author Response · Authors · 2025-08-05
> > >
> > > Dear Reviewer eadU,
> > >
> > > We sincerely thank the reviewer for your thoughtful feedback and for raising the recommendation to a borderline accept. We are glad to know that our rebuttal has helped clarify several of your concerns, and we greatly value your constructive perspective. Your feedback and suggestions are insightful and help us to improve our paper.
> > >
> > > Thank you very much,
> > >
> > > Authors

---

### Official Review · Reviewer_uDbX · 2025-07-03

**Clarity:** 4
**Significance:** 4
**Originality:** 4
**Rating:** 5
**Confidence:** 3

**Summary:**

This paper introduces MANGO framework which is for multimodal fusion that leverages normalizing flows combined with an invertible cross-attention (ICA) mechanism.  MANGO incorporates three partitioning strategies (MMCA, IMCA, and LICA), with LICA introducing a learnable token permutation via LU decomposition. Experiments across semantic segmentation, image translation, and multimodal classification tasks demonstrate state-of-the-art performance.

**Questions:**

- Could the author provide a comparative analysis of computational cost? Like a table comparing MANGO to key baselines on metrics such as total parameters, training time, and/or inference speed on a standard input size.
- Can MANGO handle generation tasks (e.g., image captioning or text generation), or is it limited to fusion-only applications?

**Ethical Concerns:**

["NO or VERY MINOR ethics concerns only"]

**Final Justification:**

I recommend Accept. The paper's core contribution is novel and the state-of-the-art results are compelling.

My primary initial concern about the lack of computational cost analysis was fully resolved by the authors' rebuttal. They provided data demonstrating that their method is not only more accurate but also computationally competitive with other leading models.

**Limitations:**

Yes

**Quality:**

4

**Strengths And Weaknesses:**

Strengths
- The paper is generally well-written, with a clear structure.
- Experimental results are compelling, showing state-of-the-art performance across multiple tasks including segmentation, image translation, and classification.
- The learnable partitioning strategy (LICA) using LU decomposition is an innovative way to address the rigidity of traditional flow partitions.

Weaknesses
- No runtime or efficiency comparison is provided. While the latent model approach is designed to improve efficiency, the overall architecture is likely to be computationally expensive compared to simpler fusion methods.
- Although the paper introduces a method to improve scalability, the authors themselves concede it remains a limitation. The complexity of the model might hinder its adoption in resource-constrained environments or for applications requiring very low latency inference.

---

> ### Author Rebuttal · Authors · 2025-07-30
>
> Dear Reviewer uDbX,
>
> We sincerely thank the reviewer for their thoughtful and constructive feedback. We are pleased that you found that **our paper is well-written with a clear structure**, and **we truly appreciate your recognition of innovative of LICA using lU decomposition and our strong experimental results**. We appreciate your constructive comments and would like to address these points as follows.
>
>
>
>
> [Q1] **Runtime Analysis**
>
>
> [A1] Our proposed approach is carefully designed with computational efficiency in mind. In particular, our multimodal latent normalizing flow approach (Section 3.3) projects high-dimensional inputs into a compact semantic space, allowing the model to avoid processing raw inputs directly and thereby reducing memory and computational cost. To address the concern about runtime, we have already provided quantitative comparisons in Table 2 (Appendix). MANGO achieves 59.2% mIoU on NYUDv2 while maintaining a competitive inference time of 144 ms, compared to 126 ms for TokenFusion and 153 ms for GeminiFusion. Similarly, MANGO uses 72.9M parameters and 152 GFLOPs, which is comparable to GeminiFusion (75.8M, 174 GFLOPs) despite offering higher performance. These results confirm that, while more expressive, MANGO remains practically efficient and scalable.
>
>
> | Method           | NYUDv2 mIOU | SUN RGB-D mIOU | PARAMS | GFLOPS | Inference Time |
> |------------------|-------------|----------------|--------|--------|----------------|
> | TokenFusion      | 54.2        | 53.0           | 45.9M  | 108    | 126 ms         |
> | GeminiFusion     | 57.7        | 53.3           | 75.8M  | 174    | 153 ms         |
> | **MANGO**        | **59.2**    | **54.1**       | 72.9M  | 152    | 144 ms         |
>
>
> [Q2] **The complexity of the model might hinder its adoption in resource-constrained environments or for applications requiring very low latency inference.**
>
>
> [A2] We have already discussed the limitations of our approach (Appendix Section B) related to scalability. This is a common challenge among expressive fusion models that aim to explicitly model high-order multimodal relationships. However, in our approach, we have taken concrete steps toward efficiency through the use of perceptual compression, which reduces input dimensionality. Invertible cross-attention layers ensure tractability while avoiding the need for dense fusion operations. As shown in Table 2 (appendix), MANGO achieves competitive inference time and computational cost (e.g., 144 ms, 152 GFLOPs) while outperforming SoTA models.
>
>
> [Q3] **Can MANGO handle generation tasks (e.g., image captioning or text generation), or is it limited to fusion-only applications?**
>
>
> [A3] We thank the reviewer for this insightful question. While MANGO is primarily designed for multimodal fusion tasks, its underlying architecture, based on invertible normalizing flows, is not inherently limited to fusion-only applications. In principle, the bijective mapping between input and latent spaces enables both inference and generation, provided appropriate task heads and decoders are used. However, this extension is beyond the scope of the current work and remains an exciting direction for future research. We leave this investigation to our future work.

---

> > ### Comment · Reviewer_uDbX · 2025-08-05
> >
> > Thanks for the authors' response. The provided performance table fully addresses my concerns about computational cost; the clarification on the scope regarding generative tasks is also reasonable. I'll maintain my "Accept" recommendation.

---

> > > ### Author Response · Authors · 2025-08-05
> > >
> > > Dear Reviewer uDbX,
> > >
> > > Thank you very much for your positive feedback and rating. We are glad to hear that your concerns have been addressed.
> > >
> > > We are grateful for your time and constructive feedback.
> > >
> > > Best regards,
> > >
> > > Authors

---

### Official Review · Reviewer_dFiC · 2025-07-05

**Clarity:** 2
**Significance:** 3
**Originality:** 3
**Rating:** 5
**Confidence:** 4

**Summary:**

This paper proposes MANGO, a novel multimodal fusion framework that combines normalizing flows with a tailored Invertible Cross-Attention (ICA) mechanism. MANGO explicitly models the joint distribution of multimodal data through invertible transformations, enabling interpretable fusion and improved performance across tasks such as semantic segmentation, image-to-image translation, and movie genre classification. It introduces three novel partitioned attention mechanisms (MMCA, IMCA, LICA) and demonstrates SoTA results on several benchmark datasets.

**Questions:**

1.	Why have you not provided attention maps or saliency maps to show how the attention mechanisms in this paper capture cross-modality correlations, validating the interpretability claim?
2.	You should clarify MANGO’s scalability and provide quantitative comparisons of computational costs against other similar and SoTA models.
3.	You should enhance architectural diagrams and include visualizations to show better how different MANGO functions work

**Ethical Concerns:**

["NO or VERY MINOR ethics concerns only"]

**Final Justification:**

The authors have properly addressed my concerns on visualizations or quantitative analyses of interpretability and scalability by adding sufficient explanations and the required visualization diagrams in the revision and appendix. However, the proof of the correctness or convergence of ICA should have been formally added to the revision. I will raise my score accordingly.

**Limitations:**

Yes.

**Paper Formatting Concerns:**

No concerns.

**Quality:**

3

**Strengths And Weaknesses:**

Strengths:
1.	Introduces a novel approach for multimodal fusion using Normalizing Flows and ICA, addressing key challenges in cross-modality correlation.
2.	Strong performance across multiple tasks (semantic segmentation, image-to-image translation, etc.).
3.	Clear implementation details and experimental setups allow for easy reproduction of results.
Weaknesses:
1.	No formal proof regarding the correctness or convergence of ICA, which weakens the paper’s theoretical foundation.
2.	Claims of interpretability are not demonstrated with concrete visualizations or quantitative analyses, leaving the interpretability claim unsubstantiated.
3.	Contradictory claims regarding scalability and lack of quantitative comparisons with other models.
4.	Visual aids are unclear, and more effective diagrams or illustrations would improve the understanding of the model.

---

> ### Author Rebuttal · Authors · 2025-07-30
>
> Dear Reviewer dFiC,
>
>
> We greatly appreciate your insightful review and valuable feedback. We are pleased you think **our proposed approach is novel and achieves the strong performance across benchmarks**. We appreciate your constructive comments and would like to address these points as follows.
>
>
> [Q1] **Formal proof regarding the correctness or convergence of ICA**
>
>
> [A1] Our proposed Invertible Cross-Attention (ICA) layer is theoretically grounded through its architectural design. In particular, as shown in Eqns. (4)–(6), ICA adopts an autoregressive formulation with an upper triangular attention matrix, ensuring invertibility due to the strictly positive diagonal entries of the softmax matrix. This structure ensures the existence of an inverse and facilitates the efficient computation of the Jacobian determinant, thereby meeting the tractability requirement of normalizing flows. Our approach follows established methods in flow-based models, such as RealNVP [1], AttnFlow [2], and Neural Autoregressive Flows [3], which similarly rely on structural properties rather than formal convergence proofs. Additionally, as in Table 6, we empirically demonstrate convergence and stability across tasks and increasing model depths. Our proposed invertible attention design is a promising direction for future work.
>
>
>
>
> [Q2] **Visualizations or quantitative analyses of interpretability**
>
>
> [A2] Our interpretability is theoretically supported through the explicit modeling of cross-modal dependencies via our normalizing flows with Invertible Cross-Attention (ICA) layers. Unlike prior implicit fusion models, ICA provides an explicit and tractable learning approach to modeling cross-modal interactions, as demonstrated in Eqns. (4)–(6), where the autoregressive formulation ensures invertibility and allows closed-form computation of the Jacobian determinant through the upper triangular structure of the attention matrix. This property enables us to trace and quantify the contribution of each modality to the final representation. Moreover, the partition-based design of MMCA, IMCA, and LICA enables modality-specific attention routing, providing a transparent mechanism for analyzing how each modality contributes to the fusion process. We further substantiate this by reporting performance gains through ablations (Tables 4–6), which quantitatively show how different attention configurations and layers affect model behavior. While figures are not permitted in the rebuttal, we refer the reviewer to Figure 1 in the appendix, which visualizes the attention maps and clearly illustrates how ICA captures modality-specific interactions (e.g., from depth to RGB). We will include additional visualization in our revised versions.
>
>
>
>
> [Q3] **Contradictory claims regarding scalability and lack of quantitative comparisons**
>
>
> [A3] We clarify that the scalability claim refers to both computational efficiency and the model’s ability to handle high-dimensional multimodal data. In particular, our approach introduces a multimodal latent normalizing flow (Section 3.3), which reduces input dimensionality via perceptual compression (Eqn. 13), thereby lowering the computational burden. This design enables the model to focus on semantic content rather than raw input redundancy. In addition, as shown in Table 2 in the appendix, we compare our model to TokenFusion and GeminiFusion in terms of parameter count, GFLOPs, and inference time. MANGO achieves higher mIoU (59.2% on NYUDv2) while maintaining comparable or better efficiency (e.g., 72.9M params vs. 75.8M in GeminiFusion, and 144 ms vs. 153 ms). It demonstrates that our approach not only scales well in practice.
>
>
> | Method           | NYUDv2 mIOU | SUN RGB-D mIOU | PARAMS | GFLOPS | Inference Time |
> |------------------|-------------|----------------|--------|--------|----------------|
> | TokenFusion      | 54.2        | 53.0           | 45.9M  | 108    | 126 ms         |
> | GeminiFusion     | 57.7        | 53.3           | 75.8M  | 174    | 153 ms         |
> | **MANGO**        | **59.2**    | **54.1**       | 72.9M  | 152    | 144 ms         |
>
>
>
>
> [Q4] **Visual aids are unclear**
>
>
> [A4] We thank the reviewer for the suggestion to improve the clarity of the visual aids. Due to the constraints of the rebuttal format, we are unable to include revised diagrams or illustrations in this response. In the revised version of the paper, we have revised and updated figures to improve readability. We also point the reviewer to Figure 1 in the appendix, which provides a concrete visualization of attention maps from the ICA layer, further supporting the interpretability and functional design of our model.
>
>
> [Q5] **Concerns about attention maps or saliency maps**
>
>
> [A5] While the main paper focuses on theoretical formulation and empirical results, we have provided qualitative evidence through attention visualizations in Figure 1 in the appendix, which illustrates how the Invertible Cross-Attention (ICA) layer captures cross-modal interactions between deep and RGB images. These visualizations support our claim that ICA explicitly models modality-aware dependencies.
>
>
> [Q6] **MANGO’s scalability and provide quantitative comparisons of computational costs**
>
>
> [A6] We provided a detailed comparison in Table 2 (Appendix) against recent SoTA models such as TokenFusion and GeminiFusion (Please refer to our answers in [A3]). MANGO achieves superior segmentation performance (e.g., 59.2% mIoU on NYUDv2) while maintaining comparable efficiency in inference time (144 ms), GFLOPs (152), and parameter count (72.9M). These results validate that MANGO scales effectively without compromising performance. We will make these points more prominent in the final version of the paper.
>
>
> [Q7] **Enhance architectural diagrams and include visualizations**
>
>
> [A7] We thank the reviewer for this helpful suggestion. We will enhance the architectural diagrams of MANGO in the final revised version.
>
>
>
>
> **References**
>
>
> [1] Dinh, et al. Density estimation using real nvp. ICLR, 2017.
>
>
> [2] Sukthanker et al. Generative flows with invertible attentions. CVPR 2022.
>
>
> [3] Huang et al. Neural autoregressive flows. ICML, 2018.

---

> > ### Author Response · Authors · 2025-08-06
> >
> > Dear Reviewer dFiC,
> >
> > We would like to thank you for your insightful feedback.
> >
> > As the deadline for the author-reviewer is approaching, we are reaching out to you to ensure that our rebuttal effectively addresses your concerns. If you have any further questions, please don't hesitate to let us know. We appreciate your invaluable input.
> >
> > Best regards,
> >
> > Authors

---

### Comment · Area_Chair_8bUb · 2025-08-04

Dear Reviewers,

The deadline for author-reviewer discussion period (August 6, AoE) is approaching soon, but some reviewers have not provided responses to the authors. Your feedback/response to the authors' rebuttal is highly expected.

Best,
AC

---

### Note · Authors · 2025-08-12

Dear ACs and Reviewers,

We sincerely thank the reviewers and area chairs for their time, thoughtful feedback, and constructive engagement throughout the review process. We are encouraged that our rebuttal has addressed several key concerns, leading to improved scores from reviewers. We especially appreciate the recognition of our paper’s clarity, technical soundness, and the strong performance of our proposed approach across multiple benchmarks.

Our work introduces MANGO, a novel Multimodal Attention-based Normalizing Flow framework that explicitly models the joint distribution of multimodal inputs via a tractable and invertible attention-based network. Unlike prior fusion methods that rely on implicit attention, MANGO offers (1) a new invertible cross-attention (ICA) layer with tractable Jacobian computation, (2) three structured partitioning strategies (MMCA, IMCA, LICA) to efficiently capture modality-specific and cross-modal relations, and (3) a Multimodal Latent Normalizing Flow that enables scalability to high-dimensional multimodal data. Our approach consistently achieves state-of-the-art performance on three diverse multimodal tasks: semantic segmentation, image translation, and movie genre classification.

We also appreciate the reviewers’ suggestions to improve the paper. We will incorporate these suggested insights into our work. We thank the reviewers again for their valuable feedback, and we hope our final clarifications support a positive recommendation.

Best regards,

Authors

---

### Decision · Program_Chairs · 2025-09-17

**Decision:**

Accept (poster)

**Comment:**

The paper proposes a Multimodal Attention-based Normalizing Flow framework that explicitly models the joint distribution of multimodal inputs through a tractable and invertible attention-based network. The paper is clearly structured and easy to follow, and it presents strong performance across multiple tasks. The authors’ rebuttal successfully addressed some concerns regarding computational cost, as well as the clarity of interpretability and scalability. These revisions should be incorporated into the final version.